# Prevalence and Risk Factors of Gestational Diabetes Mellitus in Bangladesh: Findings from Demographic Health Survey 2017–2018

**DOI:** 10.3390/ijerph19052583

**Published:** 2022-02-23

**Authors:** Tapas Mazumder, Ema Akter, Syed Moshfiqur Rahman, Md. Tauhidul Islam, Mohammad Radwanur Talukder

**Affiliations:** 1Health Research Institute, Faculty of Health, University of Canberra, Canberra 2617, Australia; tapas.mazumder@canberra.edu.au; 2Maternal and Child Health Division, International Centre for Diarrhoeal Disease Research, Dhaka 1212, Bangladesh; eakter@isrt.ac.bd (E.A.); syed.moshfiqur@kbh.uu.se (S.M.R.); 3Department of Women’s and Children’s Health, Uppsala University, MTC-huset, Dag Hammarskjölds väg 14B, SE-75237 Uppsala, Sweden; 4Health Administration, Policy and Leadership Program, Murdoch Business School, Murdoch University, Perth 6150, Australia; tauhidislam1986@gmail.com; 5Wellbeing Preventable and Chronic Disease Division, Menzies School of Health Research, Darwin 0810, Australia; 6Baker Heart and Diabetes Institute, Melbourne 3004, Australia; 7Charles Darwin University, Darwin 0810, Australia

**Keywords:** GDM, Bangladesh, prevalence, risk factors, diabetes

## Abstract

Gestational diabetes mellitus (GDM) has serious consequences for both maternal and neonatal health. The growing number of noncommunicable diseases and related risk factors as well as the introduction of new World Health Organization (WHO) diagnostic criteria for GDM are likely to impact the GDM prevalence in Bangladesh. Our study aimed to assess the national prevalence and identify the risk factors using the most recent WHO criteria. We used the secondary data of 272 pregnant women (weighted for sampling strategy) from the Bangladesh Demographic and Health Survey 2017–2018. Multivariate logistic regression was performed to determine the risk factors of GDM. The overall prevalence of GDM in Bangladesh was 35% (95/272). Increased odds of GDM were observed among women living in the urban areas (adjusted odds ratio (aOR) 2.74, 95% confidence interval (CI) 1.43–5.27) compared to rural areas and those aged ≥25 years (aOR 2.03, 95% CI 1.13–3.65). GDM rates were less prevalent in the later weeks of pregnancy compared to early weeks. Our study demonstrates that the national prevalence of GDM in Bangladesh is very high, which warrants immediate attention of policy makers, health practitioners, public health researchers, and the community. Context-specific and properly tailored interventions are needed for the prevention and early diagnosis of GDM.

## 1. Introduction

Gestational diabetes mellitus (GDM) is ‘any glucose intolerance with onset or first recognized during pregnancy’ [1]. Women with GDM are at risk of high blood pressure and, consequently, preeclampsia or eclampsia during pregnancy. Later in life, these women are also at risk of developing type 2 diabetes and cardiovascular diseases [2,3]. Moreover, the babies of mothers with GDM can suffer from macrosomia, low blood sugar, breathing problems, and type 2 diabetes that develops later in life [4,5]. GDM is believed to be multifactorial although the exact cause is yet to be established. A recent umbrella review of 30 meta-analyses reported that among 61 risk factors, the most common factors included the following: overweight or obesity, family history of diabetes, hypothyroidism, sleep-disordered breathing, and polycystic ovary syndrome [6].

Globally, the prevalence of GDM varies widely, largely because of different diagnostic criteria. The most recent meta-analysis by Saeedi et al. (2021) reported the global prevalence of GDM was 14.7% based on the International Association of Diabetes and Pregnancy Study Groups (IADPSG) criteria, the most used screening method worldwide [7]. In 2019, a meta-analysis using the same criteria reported that the highest pooled prevalence (11.4%) of GDM was in South Asia (Bangladesh, India, and Sri Lanka) compared to the rest of the world (3.6–6.0%) [8].

Few studies have reported the prevalence of GDM in Bangladesh, and most relied on the WHO (1999) criteria. The reported prevalence of GDM in these studies, largely from clinic or hospital settings, ranged between 6.8% and 40.3% [9,10,11,12,13]. The most recent estimate of GDM prevalence in Bangladesh is almost a decade old, and the prevalence was 9.7%, according to the WHO (1999) criteria. The study was limited to pregnant women visiting selected clinics for antenatal care (ANC) in 12 subdistricts of three divisions [10]. Therefore, the estimate reported in these studies does not represent the national or community prevalence. Moreover, none of these studies used the most recent WHO (2013) criteria. Factors associated with GDM reported in Bangladesh studies were advanced age, high body mass index (BMI), family history of diabetes mellitus (DM), high parities, high household income, hypertension, total year of schooling, and history of ANC [9,10,11,12,13]. However, none of these studies adjusted for covariates [9,10,11,13] or provided sufficient information about adjusted models as they reported risk factors of GDM [12].

Given the high prevalence of noncommunicable disease (NCD)-related risk factors, the high proportion of deaths claimed by NCDs [14], and the potential impact of NCDs on maternal and child health [15], it is particularly important to estimate the national prevalence of GDM based on the most up-to-date criteria. As per WHO’s estimation, NCDs were responsible for 67% of the total deaths in Bangladesh in 2016 [16]. Recent data indicates that women are at a higher risk of suffering from NCDs. According to Riaz et al., between 2010 and 2018, the proportion of Bangladeshi women with ≥3 NCD risk factors increased from 22.8% to 31.5% [17]. Therefore, it is very likely that GDM will increase as a risk, and the consequences of GDM on maternal and neonatal health outcomes will impede Bangladesh’s journey towards achieving their sustainable development goal (SDG) 3 [18]. In this context, an estimate of the national prevalence of GDM and a report on its risk factors would facilitate the development of appropriate health interventions to improve maternal and child health outcomes. Hence, our study aims to assess the prevalence and identify the risk factors of GDM in Bangladesh.

## 2. Materials and Methods

### 2.1. Study Design and Study Settings

This cross-sectional study was based on a secondary analysis of nationally representative data from the Bangladesh Demographic and Health Survey (BDHS) 2017–2018. The survey methods were published elsewhere [19]. During the BDHS 2017–2018 survey, a list of enumeration areas (EAs) was used as a sampling frame (Appendix A). An EA has around 120 households (HHs) on average and was considered the primary sampling unit (PSU) in the BDHS 2017–2018. The national survey adopted a two-stage stratified sample of HHs. Using a probability proportional sample to EA, 675 EAs (425 in rural areas and 250 in urban areas) were selected in the first stage. HHs were selected per EA using a systematic sampling approach so that it would give a statistically reliable estimation of key variables for each division, both in urban and rural areas and for the whole country [19]. This study involved a randomly selected subsample that contributed to the biomarkers’ study of all the selected HHs.

### 2.2. Study Population, Sample Size, and Sampling

To estimate the prevalence of GDM in this study, our sample included women aged 18 years or above, pregnant in any trimester, and with available data for fasting plasma glucose (FPG). In the BDHS 2017–2018, 672 clusters (EAs) and 20,160 HHs were covered. In total, 20,127 women participated in the survey. [19]. Data related to FPG were available for 7001 women aged 18 or above (87% of 8015 eligible) from a randomly selected subsample of HHs. Out of this, 6730 women were excluded as they were not pregnant during the survey. Women were asked about their pregnancy status during interviews to determine eligibility. Of the 271 pregnant women, we excluded 6 more women from the analysis because they were told either by a doctor/nurse before the current pregnancy that they had diabetes. Therefore, we were left with 265 pregnant women whose FPG was assessed, and we included their data in the analysis (Figure 1).

### 2.3. Data Collection, Outcome, and Exposure Variables

We analysed data collected from the household questionnaire, the participants’ questionnaire, and the biomarker questionnaire. More information on the questionnaire can be found in the BDHS 2017–2018 full report [19]. The data collectors requested the women not to eat or drink anything but plain water for at least 8 h prior to their visit to the households for blood glucose testing. The data collectors only measured the blood glucose level of those women who confirmed they had not taken any food or drink except for plain water for at least 8 h prior to their visit [19]. Interviewers asked study participants for their consent to collect a blood sample on the day of their interview. Appointments were made for those who gave consent, and individuals were requested to fast overnight for 8 h to take their blood sample on the following day. One male health technician and one female health technician visited the participants for the blood sample collection early the next day. Individuals failing to take the test for any reason on the scheduled day were requested again to fast overnight and take the test on the following day. The participants were visited twice in most cases [19].

### 2.4. GDM Diagnosis Criteria Used in This Study

In 2013, the WHO updated its GDM diagnosis criteria corresponding to the criteria proposed by the International Association of Diabetes and Pregnancy Study Groups (IADPSG) in 2010 [20,21]. The updated criteria include diagnosis using any of the three following values of plasma glucose:Fasting plasma glucose: 5.1–6.9 mmol/L or (92–125) mg/dLOne-hour plasma glucose: ≥10.0 mmol/L or ≥180 mg/dL following a 75 g oral glucose loadTwo-hour plasma glucose: 8.5–11.0 mmol/L or (153–199 mg/dL) following a 75 g oral glucose dose

We considered the FPG value for diagnosis of GDM in this study.

### 2.5. Variables of Interest

Explanatory variables included a set of socioeconomic and demographic characteristics: women’s age in years (<25 and ≥25) based on the median age of the study population; women’s education level (primary or below, secondary and higher); women’s occupation (employed or unemployed); place of residence (urban and rural); wealth index (lowest, middle, and highest obtained by the application of principal component analysis and based on the ownership of home goods); duration of pregnancy in weeks; hypertension (yes or no); women’s height in centimeters; and prepregnancy BMI. Prepregnancy BMI was calculated as a weight-to-height ratio (weight in kilogram/height in metre^2^). To calculate the prepregnancy BMI, we subtracted 3 kg from the current weight for women up to 20 weeks of gestation. For women who were over 20 weeks of gestation, we subtracted 3 kg plus 0.5 kg/week from their current weight [22].

### 2.6. Data Analysis

All statistical analyses accounted for the two-stage stratified sampling strategy employed in the BDHS with ‘svyset’ commands in Stata version 14.0 (Stata Statistical Software, College Station, TX, USA). Sampling weights were applied to compensate for the unequal probability of being recruited and to obtain nationally representative estimates.

We used frequency distribution to describe the socioeconomic and demographic characteristics of the mothers with and without GDM, and the chi-square test or t-test was used for comparison between groups as appropriate. We performed bivariate and multivariate analyses to determine the factors associated with GDM. We checked multicollinearity to understand if two independent variables were correlated. We also analysed the multicollinearity between residence and wealth indices. We included the variables in the multivariate analysis if they reached the 20% level of significance in bivariate analysis. In addition, the prepregnancy BMI was included in the multivariate analysis. We reported crude and adjusted odds ratio (aOR) and its corresponding 95% confidence interval (CI) from logistic regressions. We used the statistical package Stata 14 to perform the analysis.

### 2.7. Ethical Approval and Consent

This study was conducted using publicly available secondary data from the BDHS 2017–2018. The Institutional Review Boards of ICF Macro in Calverton, MD, USA, and Bangladesh Medical Research Council approved the BDHS 2017–2018. The participants were given information about the aim of the study, risks, benefits, future use of data, confidentiality, and anonymity of information. Informed consent was collected from all participants before data collection. All the identifier information was removed before we downloaded the data from the BDHS website.

## 3. Results

### 3.1. Prevalence of GDM

Table 1 presents the prevalence of GDM and sociodemographic and clinical characteristics of participants. The overall weighted prevalence of GDM in Bangladesh was 35% (95/272). Around 40% of women aged 25 years or more had GDM. A higher GDM prevalence was reported among women living in urban areas (*p* < 0.001) compared to those in rural areas, and those from the highest wealth index tended to suffer more from GDM in comparison to the other two wealth indexes (*p* = 0.025). Women with GDM were likely to be early in their pregnancy (18 weeks) compared to women in the non-GDM group (24 weeks) (*p* < 0.001). The majority of participants in the study were in their third trimester (43.6%), 37.7% were in their second trimester, and the rest (18.6%) were in their first trimester. Although less than 20% of pregnant women were in their first trimester, the diagnosis of GDM was almost 2–3 times higher among women in the first trimester group, compared to those in the second and third trimester groups (*p* < 0.001). Women with GDM had a slightly higher BMI compared to their counterparts (*p* = 0.072) (Table 1). Unweighted frequencies of sociodemographic and clinical characteristics of GDM and non-GDM participants are available in the Appendix A.

### 3.2. Risk Factors of GDM

In the multivariate adjusted model, advanced age and urban residence showed a significant association with GDM (Table 2). Women aged 25 years and above were two times (aOR 2.03, 95% CI 1.13–3.65) more likely to have GDM compared to the women aged below 25 years. Women living in urban areas were almost 3 times (adjusted odds ratio (aOR) 2.74, 95% CI 1.43–5.27) more likely to have GDM compared to those in rural areas (Table 2). The higher odds of GDM among women from the highest wealth group (unadjusted (uOR) 2.28, 95% CI 1.23–4.23) compared to that of lowest wealth group in the univariate analysis lost significance in the adjusted analysis. Women in advanced weeks of pregnancy were less likely to have GDM compared to those in early weeks of pregnancy (aOR 0.93, 95% CI 0.90–0.96).

## 4. Discussion

To the best of our knowledge, this is the first study reporting the national prevalence of GDM in Bangladesh. Our study indicates that one third of pregnant women have GDM, a rate that is 3–4 times higher than most of the previous studies in Bangladesh. The prevalence of GDM in previous studies ranged between 7% and 14%, except the study that was conducted with women visiting antenatal clinics in an urban tertiary hospital, which reported a prevalence of 40.3% as per the WHO (1999) criteria [9,10,11,12,13,23]. The prevalence reported in this study is also three times higher than the pooled overall prevalence of GDM reported in South Asian countries (Bangladesh, India, and Sri Lanka) (11.4%) [8]. An increased risk of GDM with higher age in this research is consistent with previous research [10,12,13,24].

Several factors could contribute to such a high prevalence of GDM in Bangladesh. Most importantly, we have used the most recent WHO criteria, which has a lower cutoff value of FPG (5.1–6.9 mmol/L) [1] compared to the earlier Bangladeshi studies, which used either FPG ≥7.0 mmol/L or FPG ≥5.3 mmol/L as cutoff points for the diagnosis of GDM [9,10,11,12,13]. The differential prevalence of GDM has been reported across studies globally because of different diagnostic criteria for GDM, and a higher rate was observed following the introduction of updated diagnostic criteria [8]. Second, in contrast to previous research in Bangladesh, which were mostly hospital-based studies and focused on either urban or rural populations, we have used nationally representative, population-based samples to report GDM [9,10,11,12,13,19]. Additionally, these studies varied in terms of pregnancy trimester of participants when the diagnosis for GDM was performed [9,10,11,12,13]. Evidence exists that the prevalence of GDM varies in different trimesters of pregnancies [9,24]. Lastly, an overall increase in NCD-related risk factors in Bangladesh in recent years is also likely to contribute to an increase in GDM. Riaz et al. (2020) reported almost 90% of women had inadequate fruit and vegetable consumption, and 15% had insufficient physical activity. Moreover, the prevalence of overweight and obesity was also significantly higher in women [17]. The recent BDHS report also showed an increase in the prevalence of diabetes among adult women aged 35 and older, from 12% in 2011 to 14% in 2017–2018 [19].

We found that pregnant women living in urban areas were more at risk of developing GDM compared to those living in rural areas, which is consistent with studies elsewhere [25,26]. A higher prevalence of diabetes in urban populations than that of rural populations in Bangladesh has been observed [27,28]. According to the BDHS 2017–2018 survey, the prevalence of diabetes among women 18 years and older was significantly higher in urban than rural areas (14% versus 8%) [19]. Similarly, a higher prevalence of NCD-related behavioural and biochemical risk factors was observed in urban populations. The urban population has shown a very high prevalence of inadequate vegetable and fruit consumption (92%), insufficient physical activity (14%), and overweight and obesity (34%) [17]. Further research will be pertinent to identify high-risk groups, especially within urban populations and their risk factors related to GDM to promote appropriate intervention strategies. In addition, urban and maternal health strategies in Bangladesh should consider the needs of pregnant women in the context of growing NCDs to ensure maternal and neonatal wellbeing.

Unlike other studies, our study found that rates of GDM were lower among women in advanced stages of pregnancy. A nationally representative study conducted in India reported that the age-adjusted prevalence of GDM was lower among women in the second trimester of pregnancy compared to that among women in the first trimester of pregnancy. However, the prevalence among women in the third trimester was higher than both women in the first trimester and women in the second trimester [29]. Of the Bangladeshi studies, Sultana et al. reported no significant difference in the prevalence of GDM among women in different trimesters of pregnancy [12]. Sayeed et al. reported that there was no significant difference in GDM prevalence between women less than 26 weeks pregnant and women 26 or more weeks pregnant [30]. Another hospital-based observational study reported the prevalence of GDM among the women in the 2nd and 3rd trimester were more than two times and four times higher, respectively, than that of women in the 1st trimester [9]. Screening and diagnosis of GDM is traditionally delayed until the late second or early third trimester of pregnancy to maximize GDM detection rate as pregnancy-induced diabetogenic effects increase with gestation [31]. However, because of the growing number of type 2 diabetes among women of reproductive age, screening women early in their pregnancy for abnormal blood glucose level is likely to improve pregnancy outcomes through appropriate dietary advice and pharmacological interventions [9,12]. Although the evidence regarding the benefits of early screening and treatment of GDM on pregnancy outcomes remains inconclusive, including evidence from small, randomised controlled trials, further large studies have been recommended among different population groups, including those from low- and middle-income countries and in low-risk populations [32]. Our findings add to calls for such a study to target GDM interventions in different contexts.

Our study shows several strengths. Firstly, all the studies that reported GDM prevalence in Bangladesh used older diagnosis criteria and were conducted 8–21 years ago [9,10,11,12,13]. In contrast, our study used the updated diagnosis criteria and analysed the most recent BDHS dataset, which has the statistical merit of providing national level estimates [1,19]. Secondly, in the context of rapid urbanisation, industrialisation, and dominance of sedentary lifestyles, evidence exists that the prevalence of NCDs and related risk factors are growing in Bangladesh [17,28,29]. This indicates that the prevalence of GDM is likely to increase, and our study findings are consistent with this assumption. Higher FPG levels (but below the threshold for GDM diagnosis) in early pregnancy are also associated with increased risks of later diagnosis of GDM and adverse pregnancy outcomes [1]. Thirdly, the high prevalence of GDM reported in our study and in previous studies [9,10,11,12,13] suggests that immediate maternal health interventions are required to address NCDs, including GDM. Studies have identified the limitations of Bangladesh’s health system preparedness, including a lack of adequate knowledge and training of health practitioners and managers about GDM as well as the absence of a management guideline [33]. Routine screening of all pregnant women for GDM in all levels of health facilities in Bangladesh should be undertaken to promote early interventions and prevent adverse maternal and neonatal health outcomes.

The main limitation of this study is that this is a cross-sectional study. Causal relationships cannot be established based on the data of cross-sectional studies. Another limitation is that the survey team solely depended on pregnant women’s reporting about their fasting for at least 8 h prior to blood tests and their pregnancy status. Elsewhere, a decline in FPG by about 0.5 mmol/L (9 mg/dL) by the end of the first trimester or early in the second trimester has been reported [21]. Therefore, using a FPG cutoff of 5.1 mmol/L in our research might have misclassified some GDM cases in nonobese women close to the cutoff level early in first trimester of pregnancy. The prepregnancy BMI was calculated based on a specific formula proposed in the Institute of Medicine’s guideline [22], which could be a source of error. However, this is unlikely to influence the reported prevalence of GDM in this study. We were also unable to assess the influence of some other important risk factors of GDM, such as family history of diabetes, history of macrosomia, congenital anomalies, stillbirth, etc., due to unavailability of data on these factors in the BDHS 2017–2018 dataset [19,34].

## 5. Conclusions

The national prevalence of GDM in Bangladesh is quite high and warrants the immediate attention of policy makers, health practitioners, public health researchers, and the community. Context-specific and properly tailored interventions are needed for the prevention and early diagnosis of GDM. Long-term health and economic burdens will be inevitable unless prompt actions are taken.

## Figures and Tables

**Figure 1 ijerph-19-02583-f001:**
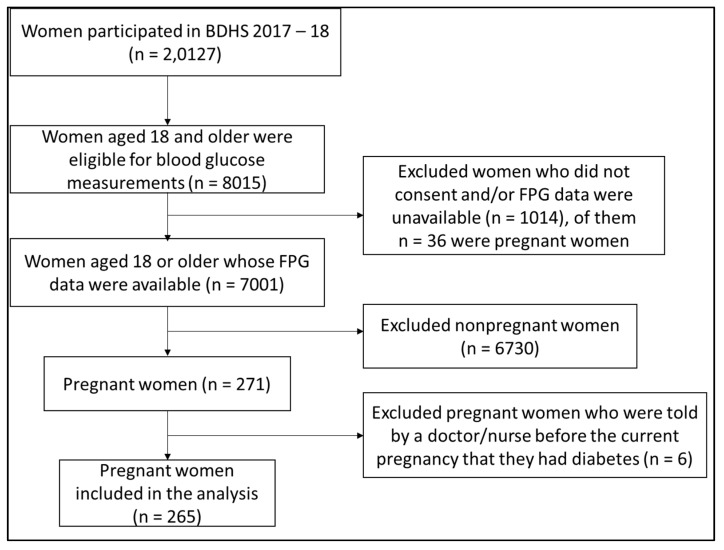
Flow of sample selection (unweighted). Abbreviation, FPG—fasting plasma glucose.

**Table 1 ijerph-19-02583-t001:** Sociodemographic and pregnancy characteristics of the study participants and prevalence of gestational diabetes mellitus.

Variables *n* (%)	Total (Weighted) Frequency of Pregnant Women *n* (%) N = 272	Women with GDM ^a^ *n* (%) *n* = 95	Women without GDM ^a^ *n* (%) *n* = 177	*p*-Value
Age group				0.145
<25	153 (56.1)	47 (30.7)	106 (69.3)	
≥25	119 (43.9)	48 (40.5)	71 (59.5)	
Education				0.350
Primary or below	105 (38.6)	31 (29.2)	74 (70.8)	
Secondary	128 (47.3)	50 (39.0)	78 (61.0)	
Higher	38 (14.1)	14 (37.3)	24 (62.7)	
Occupation				0.585
Employed	100 (36.7)	33 (32.6)	67 (67.4)	
Unemployed	172 (63.3)	63 (36.4)	110 (63.6)	
Place of residence				<0.001
Rural	196 (72.0)	54 (27.4)	142 (72.6)	
Urban	76 (28.0)	41 (54.5)	35 (45.5))	
Wealth index				0.025
Lowest	93 (34.4)	27 (28.8)	66 (71.2)	
Middle	93 (34.2)	27 (29.3)	66 (70.7)	
Highest	86 (31.5)	41 (48.0)	45 (52.0)	
Birth order				0.452
First	92 (34.0)	27 (29.3)	65 (70.7)	
Second	88 (32.6)	34 (38.7)	54 (61.3)	
Third	91 (33.4)	34 (37.1)	57 (62.9)	
Duration of pregnancy in weeks				<0.001
Mean (SD ^b^)	21.7 (9.7)	17.8 (8.9)	23.8 (9.5)	
Pregnancy trimesters ^c^				<0.001
First	49 (18.6)	33 (67.2)	16 (32.8)	
Second	100 (37.7)	36 (36.5)	63 (63.5)	
Third	115 (43.6)	23 (20.3)	92 (79.7)	
BMI ^d^				0.072
Mean (SD ^b^)	19.6 (4.1)	20.2 (4.0)	19.4 (4.2)	
Hypertension				0.310
Yes	249 (91.5)	84 (33.9)	164 (66.1)	
No	23 (8.5)	11 (46.3)	12 (53.6)	

^a^ GDM—gestational diabetes mellitus. ^b^ SD—standard deviation. ^c^ Trimesters of pregnancy—first trimester 0–12 weeks, second trimester 13–24 weeks, third trimester > 24 weeks. ^d^ BMI—prepregnancy body mass index (weight in kg/height in m^2^) (*n* = 257), calculated by subtracting 3 kg from the current weight for women up to 20 weeks of gestation and 3 kg plus 0.5 kg/week from their current weight for women above 20 weeks of gestation.

**Table 2 ijerph-19-02583-t002:** Association between women’s demographic and clinical characteristics with risk of gestational diabetes mellitus.

Characteristics	uOR ^a^ (95% CI)	aOR ^b^ (95% CI)
Age (group)		
<25	1.00	1.00
≥25	1.53 (0.93–2.54)	2.02 (1.13–3.62)
Place of residence		
Rural	1.00	1.00
Urban	3.18 (1.83–5.51)	2.74 (1.43–5.28)
Wealth quintile		
Lowest	1.00	1.00
Middle	1.02 (0.54–1.93)	0.84 (0.41–1.70)
Highest	2.28 (1.23–4.23)	1.22 (0.58–2.59)
Duration of pregnancy (week)	0.93 (0.91–0.96)	0.93 (0.90–0.96)
BMI ^c^	1.05 (0.98–1.11)	0.99 (0.92–1.96)

Table 2 legend. ^a^ uOR—unadjusted odds ratio. ^b^ aOR—adjusted odds ratio (model was adjusted for age, place of residence, wealth index, duration of pregnancy, and pre-pregnancy BMI). ^c^ BMI—prepregnancy body mass index (weight in kg/height in m^2^) (*n* = 257), calculated by subtracting 3 kg from the current weight for women up to 20 weeks of gestation and 3 kg plus 0.5 kg/week from their current weight for those over 20 weeks of gestation.

## Data Availability

The data is publicly available at the Demographic and Health Survey (DHS) website. Anyone can access the data submitting a concept note of the study to DHS.

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
