# Peer review of "Prevalence and Risk Factors of Gestational Diabetes Mellitus in Bangladesh: Findings from Demographic Health Survey 2017–2018"

_ijerph, 2022, doi:10.3390/ijerph19052583_

Round 1
Reviewer 1 Report
This article/manuscript adresses a major public health concern. Gestational diabetes is a major contributor to adverse outcomes in both mother and baby during pregnancy.
The Introduction adequately justifies the need for this study. The writing style and language throughout the manuscript is adequate.
Major concerns:
- It is not clear how the participants (subsample) were selected from the eligible population, i.e. how the 7001 participants were selected form the 20,127 women who participated in the study. Was it random selection?
- Could sensitivity analyses be performed for the reported prevalence?
- It is not reported what proportion of women did not have a FPG done, and if there is the potential that this study could have been impacted by a selection bias if the particpants were systematically different to those women who did not have FPG done.
Minor recommendations:
- Suggest including a flowsheet of the recruitment process in the initial BDHS survey 2017-18.
- Pregnancy status in this study only confirmed patient reports and not objective tests. Should be listed as a limitation.
- Pre-pregnancy BMI also calculated on a specific formula which could be a source of error, and should be listed as a limitation.
- The fasting preceding the FPG level is based on the mother's verbal report of such fasting and could not otherwise be confirmed. The authors acknowledged this limitation in the manuscript.
Reviewer 2 Report
Prevalence and risk factors of gestational diabetes mellitus in Bangladesh: findings from Demographic Health Survey 2017–2018
This study aims to assess the prevalence and identify risk-factors of GDM in Bangladesh. The study is interesting and important and allows for a broad look at the diabetes situation in Bangladesh during 2017-2018. It is emphasizes the importance of obtaining the immediate attention of policy makers, health practitioners, public health researchers and the community to manage the early prognosis of GDM in women.
A few comments:
- In the Methods and Results sections, it is not clear when the women were recruited (week/trimester).
- Was there any follow-up of the women or were data collected once on recruitment day? Unclear
- What is the reason for a cutoff age of 25? Please verify in Methods.
- Are there any updates for recent years (2018-2021)? The research covered only one year: 2017-2018.
- Abbreviations – please write the complete term the first time it is used in the paper (example (FPG page, 3 line 102). The authors should add a list of abbreviations at the beginning or end of the manuscript.
- Table 1 and Table S1- SD not sd
Round 2
Reviewer 2 Report
The text has been modified according to my suggestions.